# Metrological Qualification of PD Analysers for Insulation Diagnosis of HVDC and HVAC Grids

**DOI:** 10.3390/s23146317

**Published:** 2023-07-11

**Authors:** Fernando Garnacho, Fernando Álvarez, Alf-Peter Elg, Christian Mier, Kari Lahti, Abderrahim Khamlichi, Eduardo Arcones, Joni Klüss, Armando Rodrigo Mor, Pertti Pakonen, José Ramón Vidal, Álvaro Camuñas, Jorge Rovira, Carlos Vera, Miran Haider

**Affiliations:** 1Fundación para el Fomento de la Innovación Industrial, FFII-LCOE, Eric Kandel Street 1, Getafe, 28906 Madrid, Spain; fernando.garnacho@ffii.es (F.G.); jose.vidal@ffii.es (J.R.V.); jrovira@ffii.es (J.R.); 2Department of Electrical and Electronic Engineering, Automatic Control and Applied Physics, School of Industrial Design and Engineering (ETSIDI), Universidad Politécnica de Madrid (UPM), Ronda de Valencia 3, 28012 Madrid, Spain; fernando.alvarez@upm.es (F.Á.); eduardo.arcones@upm.es (E.A.); a.camunas@upm.es (Á.C.); carlos.vera.toluca@gmail.com (C.V.); 3RISE Research Institutes of Sweden, 501 15 Borås, Sweden; alf.elg@ri.se (A.-P.E.); joni.kluss@ri.se (J.K.); miran.haider@ri.se (M.H.); 4Delft University of Technology, Mekelweg 4, 2628 CD Delft, The Netherlands; c.mierescurra@tudelft.nl; 5Department of Electrical Engineering, Tampere University (TAU), Korkeakoulunkatu 3, 33720 Tampere, Finland; kari.lahti@tuni.fi (K.L.); pertti.pakonen@tuni.fi (P.P.); 6Instituto de Tecnología Eléctrica, Universitat Politècnica de València, Camino de Vera s/n, 46022 Valencia, Spain; arrodmor@ite.upv.es

**Keywords:** partial discharges, cable insulation, GIS, condition monitoring, insulation testing, qualification procedure, standardisation, HVDC–HVAC transmission and distribution grids, off-line measurement, on-line measurement

## Abstract

On-site partial discharge (PD) measurements have turned out to be a very efficient technique for determining the insulation condition in high-voltage electrical grids (AIS, cable systems, GIS, HVDC converters, etc.); however, there is not any standardised procedure for determining the performances of PD measuring systems. In on-line and on-site PD measurements, high-frequency current transformers (HFCTs) are commonly used as sensors as they allow for monitoring over long distances in high-voltage installations. To ensure the required performances, a metrological qualification of the PD analysers by applying an evaluation procedure is necessary. A novel evaluation procedure was established to specify the quantities to be measured (electrical charge and PD repetition rate) and to describe the evaluation tests considering the measured influence parameters: noise, charge amplitude, pulse width and time interval between consecutive pulses. This procedure was applied to different types of PD analysers used for off-line measurements, sporadic on-line measurements and continuous PD monitoring. The procedure was validated in a round-robin test involving two metrological institutes (RISE from Sweden and FFII from Spain) and three universities (TUDelft from the Netherlands, TAU from Finland and UPM from Spain). With this round-robin test, the effectiveness of the proposed qualification procedure for discriminating between efficient and inappropriate PD analysers was demonstrated. Furthermore, it was shown that the PD charge quantity can be properly determined for on-line measurements and continuous monitoring by integrating the pulse signals acquired with HFCT sensors. In this case, these sensors must have a flat frequency spectrum in the range between several tens of kHz and at least two tens of MHz, where the frequency pulse content is more significant. The proposed qualification procedure can be useful for improving the future versions of the technical specification TS IEC 62478 and the standard IEC 60270.

## 1. Introduction

Although there are still challenges to overcome for the improvement of off-line and on-line PD measurements in HVDC grids, PD measurements have proven to be very useful for detecting insulation defects in HVAC installations [1,2,3,4,5,6,7]. The most critical insulation defects that generate PD activity in electrical grids are internal or void-type defects and internal surface defects. Less consideration should be given to external surface defects or to floating potential defects and when the corona effect is detected. Among the inherent drawbacks of PD measurements in HVAC are the lack of applicable standards, absence of reference insulation diagnosis criteria, and the difficulty in detecting, separating and locating PD sources in the presence of electric noise. These difficulties are also present in HVDC with the added inconveniences of the low generation of PD pulses and high levels of electronic noise present. In HVDC, except with the corona insulation defect, PDs mainly appear when the voltage value changes, e.g., due to polarity inversion, voltage fluctuations or when a temporary or transient overvoltage occurs [8,9,10,11,12]. In addition, the existence of pulsating noise generated in the power electronic converters of the substations makes the measurement especially difficult. Monitoring applications, particularly those operating in HVDC, require efficient HFCT sensors, digital recorders and PD signal processing tools (hereinafter referred to as PD analysers) for the adequate detection of PDs that may travel distances of several kilometres until they arrive to the measuring point, where the cable sheaths are accessible [13]. In this sense, to assure adequate PD measurements and analysis, the qualification of the PD analysers is necessary. Furthermore, the establishment of a homogeneous criterion for the qualification process is also required. In on-site PD tests, special attention must be paid to the influence factors that affect the measurements. These influence factors can be summarised as follows: the background noise, the PD magnitude variability, the PD pulses width and the minimum time interval between consecutive pulses. The previous influence factors that were considered in the set of four tests proposed in this research are described in Section 5. For the qualification of PD analysers, robust measurements and appropriate requirements are considered in these tests. The stochastic behaviour associated with PD activity does not allow for the correct qualification of PD analysers when real insulation defects generated in test cells, for example, are used. For the correct performance of these tests, controllable series of PD pulses and noise signals artificially generated are required. In this research, an industrially developed portable synthetic PD calibrator was used [14]. Special requirements were established depending on the applicability of the PD analyser under qualification: for off-line PD measurements, for on-line sporadic PD measurements or for continuous PD monitoring. To perform the four metrological qualification tests proposed, several PD pulse trains and electric noise signals were generated by means of this synthetic PD calibrator, which uses an arbitrary wave generator with an impedance load of 50 Ω and 400 MHz bandwidth [14].

In Section 2, the reference PD pulse trains and noise signals used in the metrological tests are described. In Section 3, the importance of accurately knowing the pulses’ charge value is explained. Different charge measurement methods are presented in Section 4. The metrological tests are described in Section 5. The practical cases implemented in the round-robin test for the metrological qualification of PD analysers are shown in Section 6. Finally, in Section 7, the conclusions are presented.

## 2. Reference PD Pulse Trains and Noise Signals

For the realisation of the four metrological tests that are described in Section 3, reference PD pulse trains and noise signals were generated.

### 2.1. Reference PD Pulse Trains Used for the Metrological Tests

The waveform of the pulses generated in the PD trains corresponds to an inverse double exponential function (IDE) according to Equation (1).
(1)$$i\left(t\right)=i_{peak}\cdot k\cdot \frac{1}{e^{\alpha \cdot t}+e^{-\beta \cdot t}}$$
where *i_peak_* is the peak value of the current pulse *i*(*t*) and *α* and *β* are two time constants. The parameter *k* is obtained with the following equation:(2)$$k=\frac{\beta +\alpha }{\beta }\cdot {\left(\frac{\beta }{\alpha }\right)}^{\frac{\alpha }{\beta +\alpha }}$$

The charge value of this current pulse can be calculated by Equation (3):(3)$$q_{\infty }=\int_{0}^{\infty }i\left(t\right)\cdot dt=i_{peak}\cdot \int_{0}^{\infty }i_{p.u.}\left(t\right)\cdot dt=i_{peak}\cdot T_{PD}$$

*i_p.u._* is the current pulse per unit.

The PD time parameter, *T_PD_*, is defined as the width of the equivalent rectangular pulse that has the same charge, q, and current peak, $i_{peak}$, values as the original current PD pulse. The *T_PD_* value of a PD pulse waveform following an IDE function can be determined by (4) using the trigonometric function cosecant (csc):(4)$$T_{PD}=\frac{\pi }{\beta }\cdot {\left(\frac{\beta }{\alpha }\right)}^{\frac{\alpha }{\beta +\alpha }}\cdot \csc\left[\frac{\pi \cdot \alpha }{\alpha +\beta }\right]$$

An IDE pulse with a *T_PD_* of 75 ns (*T_1_*/*T_2_* = 31.2/76 ns) and a cut-off frequency of 3.3 MHz was considered as the reference PD current pulse for most of the metrological tests (see Figure 1). In all the metrological tests performed, each PD pulse belonging to a PD train was made with the reference PD pulse (*T_PD_* = 75 ns), except in the “PD time test”, in which six *T_PD_* values were used, from *T_PD_* = 8 ns to 150 ns.

A summary of the parameters defined for the pulse trains to be generated for the qualification tests is shown in Table 1. Furthermore, for all metrological tests except for the “resolution time test”, the trains were made up of consecutive bursts of four pulses of equal amplitude, generated every 10 ms (400 pulse/s) with 1 ms between them. The “linearity test” was performed generating seven PD trains with the reference PD pulse, but with different charge values from 10 pC to 2.4 nC. For the “resolution time test”, the pulses of each train were generated at equal time intervals from 2.500 μs to 10 μs. Depending on the metrological test to be performed on a specific PD analyser, a different type of noise signal must be superimposed on the PD trains. The characteristics of the noise signals used in the metrological tests are presented in the next subsection.

### 2.2. Pulsating and Non-Pulsating Noises

Extraneous signals detected during PD tests without any correlation with the PD pulses generated in the insulation media are considered by the IEC 60270 standard as background noise and interferences. These signals are normally caused by electronic devices using thyristors or IGBTs, broadcast radio disturbances, communication devices, etc. The noise signals can be classified into two types: non-pulsating noises and pulsating noises.

Non-pulsating noises, such as, for example, those related to broadcast disturbances, are usually limited to discrete frequency bands. These disturbances can affect the PD detection sensitivity if their frequency spectrum is within the measuring frequency interval of the PD analyser. To mitigate this type of noise, the gain of the instrument amplifier can be reduced using band stop filters tuned to the disturbances frequencies. On the other hand, pulsating noises can be confused with PD pulses, which implies a difficulty when making the diagnosis. In this case, to reduce the inconveniences of this type of noise, signal processing and clustering techniques must be applied. Considering this, noise rejection tools must operate in two steps. The first step should be focused on rejecting the non-pulsating noise signals by means of band stop or wavelet filters [2,15,16,17,18,19,20] and the second step should address rejecting pulsating noises by means of clustering tools [21,22,23]. The metrological tests are focused on analysing the error caused by the filtering tools when non-pulsating noise signals are rejected.

Three different types of non-pulsating noises were designed to qualify the noise filtering tools of PD analysers in the metrological tests. The first one, noise #1, was created for checking PD analysers operating in off-line measurements according to the reference standard IEC 60270 [24,25,26,27,28]. With the measuring technique proposed in this standard, a suitable frequency band can be adjusted to filter the non-pulsating noise. The second noise, noise #2, was created for checking PD analysers operating in on-line sporadic measurements according to the technical specification TS IEC 62478 [29,30]. With the measuring techniques applied in on-line measurements, generally suitable frequency bands above 1 MHz are selected, avoiding the high noise levels that are present below this frequency. The third noise, noise #3, was created for checking PD analysers operating on-line in continuous PD monitoring according to TS IEC 62478. The frequency spectrum of this noise is variable over time. This is carried out to simulate the changing behaviour of the noise signals that occurs when HVAC and HVDC installations are continually monitored. The frequency spectrum of the three noises designed is shown in Table 2.

The type and amplitude of the noise signal to be superimposed onto the PD trains for the metrological tests are specified in Table 1 and Table 2. The noise amplitudes show in Table 1 refer to the maximum amplitude of the pulses of the PD trains generated. Noise signals and PD pulse trains used in the metrological tests were generated using the synthetic PD calibrator. This calibrator injects pulsating signals (PD pulses and pulsing noises) and non-pulsing noises into a current loop [14]. The HFCT sensor of the qualifying PD analyser can be installed in the current loop, where the pulse trains can be configured with the required charge, PD repetition rate and different PD pulse widths.

## 3. Importance of the PD Charge Value and Considerations about the Estimation Method

In PD diagnosis, when a defect is detected, its level of criticality is determined by analysing, among other variables, the pulse repetition rate (PD rate) *n* and the accumulated charge quantity (*q_a_*) over time [31]. Thus, these two variables are very important for assessing the insulation condition of HV electrical installations. The charge values of the PD pulses, q, are visualised in PD patterns for HVAC and HVDC measurements and also in graphics and histograms for HVDC measurements. The accumulated charge is determined considering the charge of all pulses exceeding a specified threshold level in a considered time interval. As the charge value of a PD pulse gives valuable information to the expert analyst for performing accurate diagnosis, estimation with the accuracy of this parameter should be checked in the metrological qualification tests of PD analysers.

The PD charge measurement along a cable system is a very important parameter because the charge of PD pulses traveling through the cable sheath of a transmission HV cable system remains almost constant. When the PD pulses travel along the cable, their amplitude attenuates and their width increases almost in the same proportion, keeping its charge nearly constant. The previous statement can be demonstrated by taking into account the current pulse *i*(*t*) along the cable in the frequency domain *Ix*(*ω*):(5)$$I_{x}\left(\omega \right)=I_{o}\left(\omega \right)\cdot e^{-\gamma \left(\omega \right)\cdot x}$$
where
(6)$$\gamma \left(\omega \right)=\sqrt{\left(r+j\cdot \omega \cdot l\right)}\cdot \sqrt{\left(g+j\cdot \omega \cdot c\right)}$$

*r*, *l*, *g* and *c* are the characteristic cable parameters resistance, inductance, conductance and capacitance per unit length, respectively. Considering that the pulse charge value with the distance *x*, *q_x_*, is gisven by *Ix*(*ω* = 0), the following expression is obtained:(7)$$q_{x}=I_{x}\left(0\right)=I_{o}\left(0\right)\cdot e^{-\gamma \left(0\right)\cdot x}=q_{o}\cdot e^{-\sqrt{r\cdot g}\;\cdot x}$$

With the usual values of *r* and *g*, the value of the exponential term is close to one. For example, for the case of a pulse that has travelled ten kilometres through a transmission HVDC 320 kV cable system of 2500 mm^2^ C_u_ with *r* = 150 mΩ/km and *g* = 10^−6^ S/km, the exponential term is 0.996; see expression (8). Thus, it can be stated that the charge value of a PD pulse after travelling ten kilometres remains practically constant. Accordingly, the charge value of a PD pulse can be determined when it is measured with accuracy at any point where the cable sheaths are accessible.
(8)$$q_{10km}=q_{o}\cdot e^{-0.0039}=0.996\cdot q_{o}$$

In practice, the attenuation of the higher-frequency components makes the lower-frequency components the only ones left, but they may be below the noise, making a good estimate impossible, especially in the frequency ranges < 0.5 MHz, where the sensitivity of the HFCT is generally lower.

## 4. Charge Measurement Methods

Depending on the type of PD test (off-line measurements, on-line sporadic measurements or on-line continuous monitoring), the PD pulse is measured using different measuring methods and sensors. In off-line measurements with low levels of background noise, the conventional method based on the reference standard IEC 60270 can be applied. In this case, PD activity is measured using a quadrupole and the pulse charge is determined by applying the quasi-integration method [24,27]. However, when PD measurements are performed on-line in sporadic tests or in continuous monitoring systems, the conventional method is not suitable. This is due to the high levels of noise present in most cases in these measurements for the measuring frequencies specified in the IEC 60270 method (≤1 MHz). To achieve an adequate sensitivity in on-line PD measurements in HV grids, non-conventional methods, measuring above 1 MHz according to the technical specification TS IEC 62478, are applied. In this case, PD activity is usually measured using HFCT sensors installed in the grounding of HV cable systems, and the pulse charge can be determined by applying one of the three following methods.

### 4.1. Charge Estimation Method by Direct Reading

With this method, the voltage peak reading at the HFCT sensor output is considered and the following equation can be applied:(9)$$q\;\left(pC\right)=\frac{T_{PD}\left(ns\right)}{Z_{s}\left(\frac{mV}{mA}\right)}\cdot \;u_{peak}\left(mV\right)$$

In this approach, the charge value, *q*, depends on the pulse width (*T_PD_*), which is well known in many cases. In addition, it also depends on the frequency spectrum of the sensor transfer impedance Zs, which should remain constant up to the PD pulse cut-off frequency. The *T_PD_* of the PD pulse can be estimated using the voltage signal at the output of the HFCT sensor if its transfer function does not provoke significant signal distortion.

### 4.2. Charge Estimation Method by Applying the Quasi-Integration Approach

When a bandpass filter, defined by its lower and upper cut-off frequencies, is applied to measure PD pulse charge, the peak value of the output voltage is proportional to the charge value:(10)$$u_{f\_peak}=2\cdot \left(f_{2}-f_{1}\right)\cdot Z_{s}\cdot q$$

This approach assumes that the flat part of the pulse frequency spectrum *I(s)* remains constant in the interval defined by the cut-off frequencies *f*_1_ and *f*_2_. If the requirements of the standard IEC 60270 are met, the uncertainty for the apparent charge is less than ±10%. The charge error is larger as the upper frequency limit *f*_2_ increases because the pulse frequency spectrum cannot remain constant for very high frequencies. Figure 2 shows the frequency spectrum of two pulses, one with *T_PD_* = 150 ns and the other with *T_PD_* = 30 ns. The charge errors versus the pulse width (*T_PD_*) are analysed in Figure 3 after applying the two bandpass filters shown in Figure 2. The first filter, filter A, works in a frequency range below 1 MHz (*f*_1_ = 610 kHz, *f*_2_ = 770 kHz) and the second filter, filter B, works above 1 MHz (*f*_1_ = 1.75 MHz, *f*_2_ = 3.25 MHz). If the *T_PD_* error influence is calculated as half the difference between the maximum charge error value and the minimum charge error value, (see Equation (11)), the resulting measurement *T_PD_* error $\left(\varepsilon_{T_{PD}}\right)$ is within ±4% for the filter A and within ±25.2% for the filter B.
(11)$$\varepsilon_{T_{PD}}=\left[\mathrm{Max}\;\varepsilon -\;\min\;\varepsilon \right]/2$$

### 4.3. Charge Estimation Method by Integrating the Current Signal

Another effective method is based on the integration of the current signal in the time domain or in the frequency domain, applying Equations (12) and (13), respectively. In this case, the measured PD pulse must be reconstructed using the pulse voltage signal at the sensor output and the HFCT frequency spectrum:(12)$$q=\int_{0}^{\infty }i\left(t\right)\cdot d\tau$$
(13)$$q=I\left(\omega =0\right)$$

The integration in the frequency domain is the most accurate approach, but it is more difficult to apply. This is because it requires the characterisation of the HFCT sensor to determine the frequency spectrum of its transfer impedance and to perform complex signal processing using the raw signal data of the PD. If the transfer function of the HFCT sensor, Z_HFCT_ is very flat for the entire PD pulse frequency spectrum, waveform reconstruction is not needed. This charge estimation method can give very acceptable results as shown in Section 6.

## 5. Metrological Tests Description

For the metrological qualification of PD analysers, four tests were proposed. These tests were performed using the synthetic calibrator [14]. Their realisation enables the detection of measuring errors caused by PD analysers due to different influence parameters. 

Before the performance of the metrological tests, the PD analyser to be checked must be adjusted and calibrated. In the adjustment process, the analyser cut-off frequencies must be selected to perform the measurements in the frequency band less affected by the noise signals. This process is carried out by generating a train of reference pulses of 500 pC superimposed to the noise signal to be used in the metrological tests (noise type #1, #2 or #3). Then, the analyser must be calibrated with a train of reference PD pulses of 200 pC without the noise signal.

The first test, called noise rejection, was carried out to determine the error values in the charge (*q*) and PD repetition rate (*n*) values when various conditions of non-impulsive background noise are superimposed. The second and third tests, called linearity and PD time, respectively, were performed to determine the error values of the charge quantity for different PD pulse amplitudes and widths, respectively. The fourth test, called resolution time, was carried out to determine the PD rate error value of a PD analyser under qualification as a function of the PD repetition frequency (N).

### 5.1. Noise Rejection Test

A PD pattern related to an insulation defect is more difficult to detect and recognise the greater the noise influence in the measurement is. Noise filtering techniques are applied to improve the sensitivity in the detection of PD pulses. However, when the filtering technique is not efficient enough, some pulses are not properly acquired, making the detection and recognition of the insulation defects difficult. The objective of this test is to determine the charge and PD repetition rate error values made under a specific kind of noise, considering the PD analyser applicability, off-line tests, on-line sporadic tests or on-line continuous monitoring. For the evaluation of the noise filtering techniques used by the PD analysers to be qualified, a train of reference pulses with a constant amplitude of 100 pC ± 20% was generated. Furthermore, the three types of noises (#1, #2 or #3) presented in Table 2 were superimposed to the pulses train, according to the PD analyser applicability. The magnitude of the noise signals was adjusted in four successive steps with respect to the charge value of the pulses generated. The noise magnitudes were identified as (a) “very high noise severity” with a magnitude of 200 pC, 200% of the reference value, (b) “high noise severity” with a magnitude of 100 pC, (c) “medium noise severity” with a magnitude of 50 pC and (d) “low noise severity” with a magnitude of 20 pC. The noise magnitude was selected in decreasing order, from the “very high noise severity” to the “low noise severity”. The charge and PD rate measurement errors were determined for each noise level.

**Maximum acceptance errors for the charge and PD repetition rate values.** A maximum uncertainty for the apparent charge of ±10% is established in IEC 60270, when all requirements are met, and the noise level is lower than 50% of the pulses charge. On the other hand, in the future IEC 60270 (currently under revision), an acceptance error for the PD repetition rate, n, within ±2% is specified. These two acceptable errors are considered when a PD analyser is qualified for off-line measurements working according to IEC 60270. However, when a PD analyser is qualified for on-line sporadic tests or continuous monitoring, as, in these cases, the noise levels are higher, the maximum acceptance errors can be increased. Thus, for on-line PD measurements and continuous PD monitoring, the maximum acceptance errors established for the charge and PD repetition rate values are ±30% and ±2%, respectively, when the qualification tests are performed with “very high noise severity”.

### 5.2. Linearity Test

Errors in the scale factor linearity cause distortions in the PD patterns obtained in HVAC and HVDC measurements and also in the charge graphics and histograms in HVDC measurements. The aim of this test was to determine the scale factor linearity error made under a specific kind of noise, considering the PD analyser applicability, off-line tests, on-line sporadic tests or on-line continuous monitoring. To determine the linearity of a PD analyser, seven pulse trains were generated. All the reference PD pulses in each train had a known charge value within a tolerance of ±20%. The charge levels for the seven trains were 2.4 nC ± 20%, 1.0 nC ± 20%, 500 pC ± 20%, 200 pC ± 20%, 100 pC ± 20%, 50 pC ± 20% and 10 pC ± 20%. Furthermore, according to the PD analyser applicability, the noises #1, #2 or #3 were superimposed onto the pulse trains. Depending on the applicability of the PD analyser, the amplitude of the superimposed noises was chosen according to the percentages shown in Table 1. The linearity error influence (ε_l_) was calculated as half the difference between the maximum charge error value and the minimum charge error value. The linearity error was determined for two charge ranges, from 50 pC to 2.4 nC and from 10 pC to 2.4 nC.

**Maximum acceptance errors for the scale factor linearity.** In the standard IEC 60270, it is indicated that the maximum deviation of the scale factor k must be lower than ±5%. This maximum acceptance error, valid for PD analysers used in off-line tests, has also been adopted for PD analysers used in on-line sporadic tests or in continuous monitoring.

### 5.3. PD Time Test

The aim of this test was to determine the charge error value made when pulses with widths (*T_PD_*) are measured considering the PD analyser applicability. This charge error was determined by ºapplying six pulse trains. All PD pulses in each train had a charge value of 200 pC but differ in the pulse width, *T_PD_* = 8 ns, 16 ns, 37.5 ns, 75 ns, 110 ns and 150 ns. Depending on the applicability of the PD analyser, the amplitude of the superimposed noises was chosen according to the percentages shown in Table 1. The *T_PD_* error influence was calculated as half the difference between the maximum charge error value and the minimum charge error value. The *T_PD_* error $\varepsilon_{T_{PD}}$ was determined in two *T_PD_* ranges, from 37.5 ns to 150 ns and from 8 ns to 150 ns.

**Maximum acceptance error for the PD time (*T_PD_*)**. The maximum acceptance *T_PD_* error considered for PD analysers working in off-line tests according to IEC 60270 is ±10% because this figure corresponds to the uncertainty established in IEC 60270. For PD analysers working in sporadic on-line measurements or in continuous PD monitoring applications, a maximum acceptance error of ±30% has been established.

### 5.4. Resolution Time (t_res_) Test

When consecutive pulses are generated with very short time intervals between them, significant PD charge and repetition rate errors can arise. These errors are directly related to the resolution time of PD analysers. They also affect the ability to discriminate pulses generated in more than one source.

The objective of this test was to determine the resolution time (*t_res_*) of a PD analyser, measuring with a specific type of noise depending on the PD analyser applicability (off-line tests, on-line sporadic tests or continuous monitoring). The resolution time was determined by generating seven pulse trains. All the reference pulses in each train had a charge value of 200 pC. The trains differed in the time interval between consecutive pulses, which were 2500 μs, 320 μs, 160 μs, 80 μs, 40 μs, 20 μs and 10 μs. Depending on the applicability of the PD analyser, the amplitude of the superimposed noises was chosen according to the percentages shown in Table 1.

**Maximum resolution time value.** IEC 60270 states 5 μs to 20 µs as typical resolution time figures for a PD measuring system. For a resolution time to be accepted as a valid figure, the recorded number of pulses as observed during a defined time interval must be within ±2% of the known number of calibration pulses applied. A resolution time of at least 20 μs is required for any PD analyser.

## 6. Practical Cases of Metrological Qualification of PD Analysers

For the practical validation of the developed procedure for the metrological qualification of PD analysers, various PD analysers were qualified in a European round-robin test arranged in the framework of the project Future Energy of EURAMET. The participants were two metrological institutes RISE (Sweden) and FFII (Spain) and three European universities TUDelft (The Netherlands), TAU (Finland) and UPM (Spain). The three types of qualifications were performed on different PD analysers: off-line measurements, on-line sporadic measurements or continuous PD monitoring. The results obtained are shown in the next subsections. A synthetic PD calibrator [14] was used sequentially by all the participants to perform the metrological qualification tests presented in Section 5.

### 6.1. Metrological Qualification of PD Analysers Operating for Off-Line Measurements

Three different PD analysers operating for off-line measurements were qualified (A-1, A-2 and A-3). All of them operate according to the wideband method with an upper frequency limit f_2_ ≤ 1 MHz. One of them applies an upper frequency limit f_2_ < 500 kHz in accordance with the current IEC 60270. The other two operate with f_2_ > 500 kHz according to the new revision of IEC 60270. The lower and upper frequency limits of each PD analyser are shown in Table 3. The results of these qualification tests are also shown in Table 3. To carry out metrological tests, the non-pulsating noise #1 was superimposed on the generated pulse trains.

**In the noise rejection test**, the PD analyser A-3 meets the maximum acceptance errors considered for the charge value (Max ABS ε_q_ ≤ 10%) and for the PD repetition rate (Max ABS ε_n_ ≤ 2%), with 50% of noise #1 level. These good results were mainly due to the appropriate selection of the lower and upper frequency limits *f*_1_ and *f*_2_. The wideband measurement was tuned to a frequency range where the amplitude of the noise frequency spectrum was lower. The results obtained with A-2 and A-3 PD analysers were very affected by the noise signal due to the measurements being performed in a fixed wideband, where the noise spectrum amplitude was significant. As the noise signal #1 is high for frequencies below 500 kHz (see Table 2), the errors increase the lower the values of *f*_1_ and *f*_2_ are. Therefore, the results obtained for A-1, with *f*_1_ = 40 kHz and *f*_2_ = 400 kHz, are more affected by the noise signal.

**In the linearity test**, the PD analyser A-3 also meets the maximum acceptance error in the two charge ranges considered. The larger errors obtained for A-1 and A-2 are also affected by the noise signal.

**In the PD time test**, all the analysers meet the maximum acceptance error in the *T_PD_* range considered. The *T_PD_* errors ($\varepsilon_{T_{PD}}$) for the three PD analysers were 2.5% ± 0.1%; this figure is consistent with the theoretical error estimated in Section 4.2 (see Figure 3) for this type of measurements, which was ±4%.

**In the resolution time test**, the best result is obtained for the A-3 analyser, with a resolution time of 10 µs, for which the PD repetition rate error was 1.3%, lower than the maximum acceptable error of 2%.

In conclusion, the PD analyser A-3 is the only one that meets all the maximum acceptable errors (10% for charge measurements and 2% for PD repetition rate measurements). PD analysers A-2 and A-3 are very affected by the noise.

### 6.2. Metrological Qualification of PD Analysers Operating in On-Line Sporadic Measurements

Four different PD analysers operating for on-line sporadic measurements were qualified (B-1, B-2, B-3 and B-4). All of them operate with an upper frequency limit f_2_ > 1 MHz according to the TS IEC 62478. With each PD analyser, a different filtering method was applied (see Table 4). To carry out metrological tests, the non-pulsating noise #2 was superimposed on the generated pulse trains.

For the PD analysers B-1 and B-4, the lower and upper cut-off frequencies (*f*_1_ and *f*_2_) of the digital passband filter as well as the trigger threshold level were selected manually. Hardware filters were used in the B-2 and B-3 PD analysers. The one used by the B-2 PD analyser was a fourth-order bandpass Butterworth filter and the one used with B-3 was an eighth-order lowpass 20 MHz filter. The results of these qualification tests are also shown in Table 4.

**In the noise rejection test**, the PD analysers B-1, B-2 and B-4 meet the maximum acceptance error considered for the charge value (Max ABS ε_q_ ≤ 30%), with 200% of the noise level “very high noise severity”. Furthermore, all the PD analysers meet the maximum acceptance error considered for the PD repetition rate (≤2%), also with 200% of the noise level.

**In the linearity test**, the PD analysers B-1, B-2 and B-4 meet the maximum acceptance error in the charge range from 50 pC to 2400 pC and analysers B-1 and B-4 also meet the maximum acceptance error in the charge range from 10 pC to 2400 pC. The larger errors obtained for AB-2 and AB-3, especially in the range from 10 pC to 2400 pC, were due to the noise influence.

**In the PD time test**, all PD analysers meet the maximum acceptance error in the *T_PD_* range from 37.5 to 150 ns considered. The maximum *T_PD_* error ($\varepsilon_{T_{PD}}$) was ±29.6%, which is slightly lower than the maximum permitted error (30%). The *T_PD_* errors ($\varepsilon_{T_{PD}}$) obtained with B-1 and B-4 (±29.1% and 29.6%, respectively) are consistent with the theoretical one estimated in Section 4.2 for these measurements, which was ±25.1%. The lowest *T_PD_* error was achieved with the B-3 PD analyser, which determines the charge value by the integration of the acquired pulses in the time domain. With the B-2 PD analyser, reasonable results were achieved, but the readings were corrected with formula (1) using nominal *T_PD_* values. For this reason, the results obtained with this analyser cannot be considered as independent measurements.

**In the resolution time test**, the analysers B-1, B-2 and B-4 meet the minimum resolution time required of 20 μs. The best result of 10 µs was achieved using the analysers B-1 and B-4. A repetition rate error of 0% was obtained with analyser B-4 (see Table 4).

### 6.3. Metrological Qualification of PD Analysers for Continuous PD Monitoring

The same first three PD analysers used for online sporadic measurements (B-1, B-2 and B-3) along with a new one, C, were tested as PD analysers used for continuous monitoring. To carry out metrological tests, the non-pulsating noise #1 was superimposed onto the generated pulse trains.

The upper and lower frequency limits of B-1 and B-4 were tuned for a better charge measurement, whereas the filters of PD analysers B-2 and B-3 were the same previously used in sporadic measurements (see Table 5). With the new PD analyser “C”, an automatic wavelet filter was used for the noise rejection test, and the quasi-integration approach was applied for charge measurements. The results of these qualification tests are also shown in Table 5.

**In the noise rejection test**, the results obtained for all PD analysers are very similar to those obtained with the on-line sporadic measurements (Section 6.2). All PD analysers except for B-3 meet the requirements.

**In the linearity test**, as occurred with the on-line sporadic measurements, the most significant errors were obtained for analysers B-2 and B-3. All PD analysers meet the linearity requirements for the charge range from 50 pC to 2400 pC but only the PD analysers B-1 and B-4 meet the requirements for the charge range from 10 pC to 2400 pC.

**In the PD time test**, the maximum *T_PD_* error ($\varepsilon_{T_{PD}}$) was ±29.8%, which is slightly lower than the maximum permitted error (30%). PD analysers B-2 and B-3 are more affected by the noise signal, especially when low charge values are measured (~20 pC). However, as in the case of on-line sporadic PD measurements, the lowest *T_PD_* error was achieved with B-3. This analyser determines the charge value integrating the pulses in the time domain. The B-2 PD analyser achieved reasonable results, but the readings were corrected with formula (3) using the nominal *T_PD_* values. For this reason, the results obtained with this analyser cannot be considered as independent measurements.

**In the resolution time test**, as in the case of on-line sporadic measurements (Section 6.2), the analysers B-1, B-2 and B-4 meet the minimum resolution time required of 20 μs. The best result of 10 µs was achieved using the analysers B-1 and B-4 (see Table 5).

In conclusion, for continuous PD monitoring, the analyser B-1 and C are the only two ones that meet all the maximum acceptable errors (30% and 2% for charge and PD repetition rate measurements, respectively). However, charge errors can be improved by signal integration similar to the approach applied by analyser B-3. With the analyser B-2, the *T_PD_* value should be calculated by itself to meet all the requirements. The PD analyser B-3 is very affected by noise.

## 7. Conclusions

A procedure for the metrological qualification of PD analysers was proposed and tested, determining measurement errors caused by the influence parameters. A set of metrological tests was defined along with specific acceptance requirements. These tests were defined considering all possible applications of PD analysers, which were classified into three categories: off-line PD measurements, sporadic on-line PD measurements and continuous PD monitoring. The measurement conditions and acceptance requirements were adapted to the type of application of the analyser. The allowable errors for each test were selected considering the IEC 60270 standard requirements, the Technical Specification IEC 62478 and the inherent noise conditions of off-line and on-line sporadic measurements and continuous monitoring. For the practical validation of the qualification procedure, several PD analysers were characterised in a European round-robin test within the framework of a EURAMET research project.

For PD analysers operating in off-line measurements, the requirements related to the noise rejection test (Max |ε_q_| ≤ 10% and Max |ε_n_| ≤ 2% with 50% noise signal) were met if the cut-off frequency limits were set within a bandwidth according to the future IEC 60270 (1 MHz ≥ f_2_ ≥ 500 kHz), where the signal-to-noise ratio is larger.

PD analysers used in on-line sporadic measurements or in continuous monitoring must operate with cut-off frequencies above 1 MHz to the influence of noise signals as much as possible, since these usually have large amplitude levels below 1 MHz. In sporadic on-line measurements, the cut-off frequencies adjustment can be manual, as no significant changes are expected in the noise frequency spectrum during a PD measurement. However, in continuous PD monitoring, as the noise behaviour can change over time, an automatic filtering tool should be used, such as an automatic wavelet filter. When on-line PD sporadic measurements are performed, the selection of the cut-off frequencies of the passband filter has a significant influence on the results.

The realisation of the round-robin test was useful for the validation of the proposed qualification procedure. The results of the round-robin test show that the proposed acceptance requirements can be achieved if some technical aspects are considered; for example, the adequate selection of the cut-off frequencies, the use of automatic noise filtering tools or the appropriate current signal integration for the charge value determination. The reference qualification procedure proposed will be useful for improving the features of PD analysers and supporting the development of the future versions of PD standards.

To achieve a complete qualification of PD analysers, the metrological tests presented in this research should be complemented with diagnosis tests for diagnostic tools. In this sense, the authors have developed another procedure, also within the scope of this EURAMET project, for the qualification of PD analysers diagnostic tools (PD recognition, PD clustering and PD location). This procedure will be presented in a future paper.

## Figures and Tables

**Figure 1 sensors-23-06317-f001:**
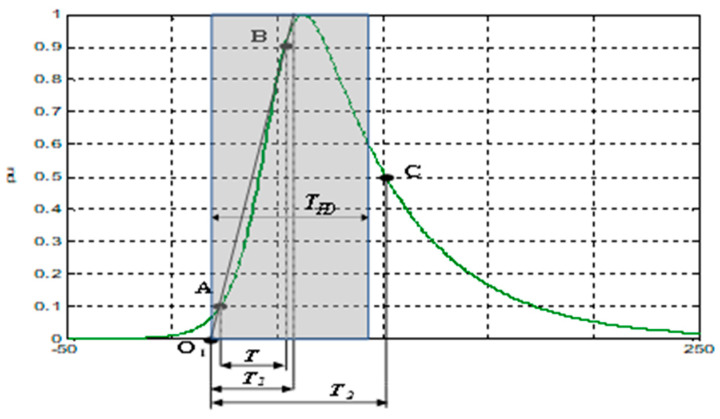
Reference IDE pulse expressed in per unit with a *T_PD_* of 75 ns, with 1/α = 44 ns and 1/β = 9.9 ns. Green line corresponds to the current pulse waveform expressed in per unit. Blue frame shaded in with grey colour represents the equivalent rectangular PD pulse with the same charge value in per unit as the original PD pulse (*T_PD_*).

**Figure 2 sensors-23-06317-f002:**
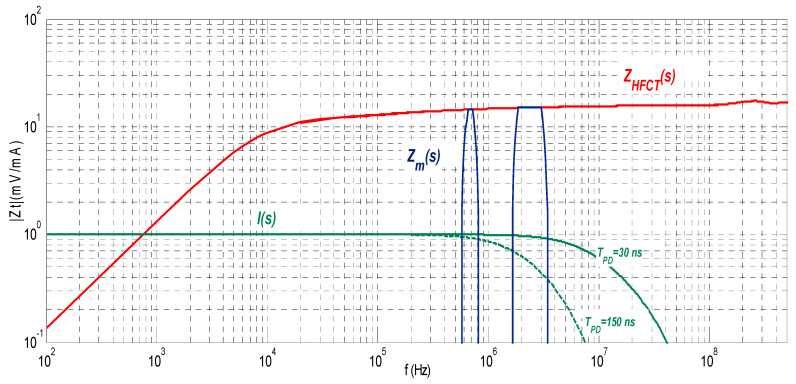
Frequency spectrum of two pulses with *T_PD_* = 150 ns and *T_PD_* = 30 ns, respectively. Bandpass filter A (frequency range below 1 MHz, *f*_1_ = 610 kHz and *f*_2_ = 770 kHz). Bandpass filter B (frequency range above 1 MHz, *f*_1_ = 1.75 MHz and *f*_2_ = 3.25 MHz). Transfer impedance of an HFCT sensor *Z_HFCT_*(*s*).

**Figure 3 sensors-23-06317-f003:**
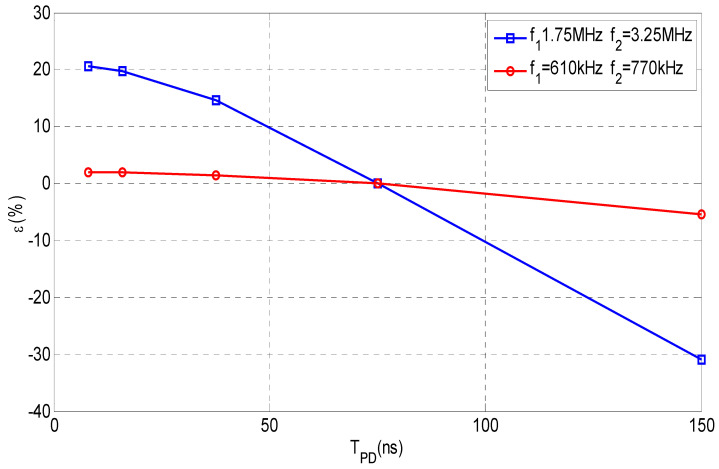
Theoretical charge errors versus pulses with different *T_PD_* resulting of applying the bandpass filter A (*f*_1_ = 610 kHz and *f*_2_ = 770 kHz), red line, and the bandpass filter B (*f*_1_ = 1.75 MHz and *f*_2_ = 3.25 MHz), blue line.

**Table 1 sensors-23-06317-t001:** Summary of the main parameters of the PD pulse trains and noises used in the metrological tests.

Test Parameters of the PD Trains	Metrological Tests for Qualification of PD Analysers			
	5.1) Noise Rejection	5.2) Linearity	5.3) PD Time (*T_PD_*)	5.4) Resolution Time (*t_res_*)
*T_PD_* (ns)	75	75	8; 16; 37.5; 75; 110; 150	75
n, N (pulse/s)	n = 400	n = 400	n = 400	N = 400; 3125; 6250; 12,500; 25,000; 50,000 and 100,000
Δt (μs)	1000	1000	1000	2500, 320, 160, 80, 40, 20 and 10 μs
Charge value *q* (pC)	100	10; 50; 100; 200; 500; 1000; 2400	200	200
Noise Amplitude	20%; 50%; 100%; 200%	35% for noises #1 and #2 and 10% for noise #3 ^(1)^		

^(1)^ The non-pulsating noise #3 amplitude (10%) is considered as lower than the one of noises #1 and #2 (35%) because it is assumed that, in continuous PD monitoring, there will be time periods with lower noise conditions than in a sporadic PD measurement.

**Table 2 sensors-23-06317-t002:** Frequency spectrum of the three types of non-pulsating noises designed for the metrological tests.

Noise #1	Noise #2	Noise #3	
Fixed noise spectrum applicable to PD analysers for off-line PD measurements	Fixed noise spectrum applicable to PD analysers for sporadic on-line PD measurements	Variable noise spectrum applicable to PD analysers for on-line PD continuous monitoring ^(1)^	
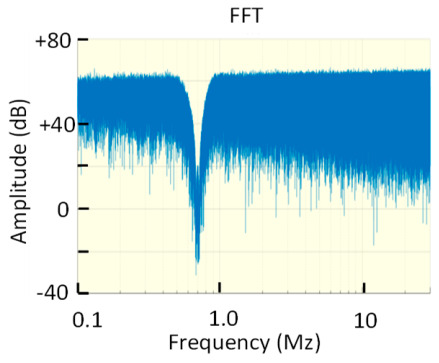	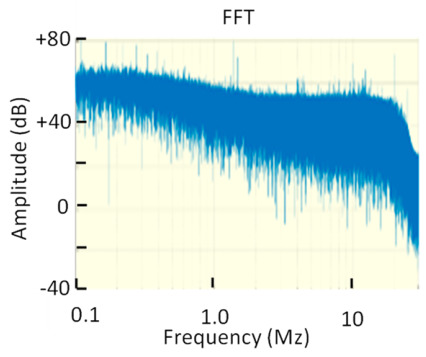	From t_n_ to t_n+1_ = t_n_ + ΔT	From t_n+1_ to t_n+2_ = t_n+1_ + ΔT
		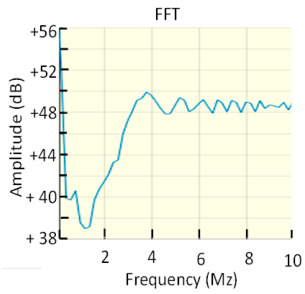	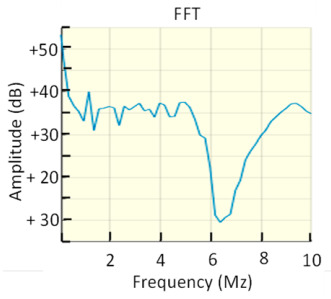

^(1)^ The frequency spectrum of noise #3 changes every time interval ΔT during the whole test time.

**Table 3 sensors-23-06317-t003:** Metrological test results of three PD analysers for off-line PD measurements under noise #1.

PD Analyser for Off-Line Measurements		Analyser A-1	Analyser A-2	Analyser A-3
f_1_–f_2_ (kHz)		40–400	40–800	610–770
Δf (kHz)		360	760	160
Charge measurement method		Quasi-integration	Quasi-integration	Quasi-integration
Filtering method		None	Manual selection of f_1_, f_2_ and	trigger level
**Metrological tests**				
**Noise** **rejection** q = 100 pC	Noise amplitude **(%)**	**Charge and PD repetition rate errors versus noise level** **Requirements** **Max** **|****ε_q_****|****≤ 10%; Max ****|****ε_n_****|** **≤ 2% under 50% noise level**		
	200	ε_q_ = 248%	ε_q_ = 151%	ε_q_ = 61.9%
		ε_n_ > 1000%	ε_n_ > 1000%	ε_n_ = −0.4%
	100	ε_q_ = 104.3%	ε_q_ = 53.9%	ε_q_ = 32.0%
		ε_n_>1000%	ε_n_ > 1000%	ε_n_ = −0.4%
	**50**	**ε_q_ = 27%**	**ε_q_ = 26%**	**ε_q_ = 2.6%**
		**ε_n_ > 1000%**	**ε_n_ > 1000%**	**ε_n_ = −0.4%**
	20	ε_q_ = 7.8%	ε_q_ = 9.6%	ε_q_ = 0.5%
		ε_n_ > 1000%	ε_n_ = 496%	ε_n_ = −0.4%
**Linearity** under 35% noise	**Charge range (pC)**	**Linearity error = ε_l_ = (Max** **ε** ** _q_ ** **value–Min** **ε** ** _q_ ** **value)/2** **Requirement ε_l_ ≤ 5% under 35% noise level**		
	50 to 2.400	±16.3%	±6.2%	±2.9%
	10 to 2.400	>100%	±6.2%	±2.9%
**PD time (*T_PD_*)** under 35% noise q = 200 pC	**Pulse width *T_PD_* range (ns)**	** *T_PD_* ** ** $\;\mathrm{error}=\varepsilon_{T_{PD}}$ ** **= (Max** **ε** ** _q_ ** **value–Min** **ε** ** _q_ ** **value)/2** ** $\mathrm{Requirement}\;\varepsilon_{T_{PD}}$ ** ** ≤ 30% under 35% noise level**		
	8 to 150	±2.5%	±2.5%	±2.7%
**Resolution time** under 35% noise q = 200 pC	** *t_res_* ** **range (μs)**	**PD repetition rate error** **Requirement Max** **|** ** *ε_n_* ** **|** **≤ 2% under 35% noise level, at least 20 μs**		
	2.500–10	2.5 ms; |*ε_n_*| > 2.0%	2.5 ms; |*ε_n_*| > 2.0%	10 μs; |*ε_n_*| = 1.3%

**Table 4 sensors-23-06317-t004:** Metrological test results of four PD analysers operating in on-line sporadic measurements under noise #2.

PD Analyser for Sporadic Online Measurements		Analyser B-1	Analyser B-2	Analyser B-3	Analyser B-4
HFCT bandwidth f_1_–f_2_ (MHz)		0.2–20	0.08–61	0.004–1.112	0.2–20
Charge measurement method f_1_–f_2_ (MHz)		Quasi-integration f_1_ = 1.75, f_2_ = 3.25	Applying formula (3)	Integration in time domain	Quasi-integration f_1_ = 2.45, f_2_ = 3.95
Digitiser: Bandwidth (Sampling rate)		30 MHz (60 MS/s)	50 MHz (100 MS/s)	20 MHz (1.25 GS/s)	50 MHz (100 MS/s)
Filtering method f_1_–f_2_ (MHz)		Passband filter Manual selection of f_1_, f_2_ and trigger level f_1_ = 1.75, f_2_ = 3.25	4th-order Passband filter Butterworth f_1_ = 0.05, f_2_ = 45 (software)	8th-order lowpass filter f_2_ = 20 (hardware)	Passband filter Manual selection of f_1_, f_2_ and trigger level f_1_ = 2.45, f_2_ = 3.95
Metrological tests					
**Noise** **rejection** q = 100 pC	**Noise** **amplitude (%)**	**Charge error and PD repetition rate errors versus noise level** **Requirements** **Max** **|****ε_q_****|**** ≤ 30%, Max** **|****ε_n_****|**** ≤ 2% under 200% noise level**			
	**200**	**ε_q_ = 8.0%**	**ε_q_ = −0.9%**	**ε_q_ = −36.5%**	**ε_q_ = 16.4%**
		**ε_n_ = 0.0%**	**ε_n_ = 0.0%**	**ε_n_ = 0.7%**	**ε_n_ = 0.0%**
	100	ε_q_ = 3.2%	ε_q_ = −0.4%	ε_q_ = 13.0%	ε_q_ = 8.1%
		ε_n_ = 0.0%	ε_n_ = 0.0%	ε_n_ = 0.7%	ε_n_ = −0.0%
	50	ε_q_ = −0.9%	ε_q_ = −0.2%	ε_q_ = 10.4%	ε_q_ = 3.3%
		ε_n_ = 0.0%	ε_n_ = 0.0%	ε_n_ = 0.7%	ε_n_ = −0.0%
	20	ε_q_ = 0.2%	ε_q_ = −0.1%	ε_q_ = 1.7%	ε_q_ = 1.0%
		ε_n_ = 0.0%	ε_n_ = 0.0%	ε_n_ = 0.7%	ε_n_ = −0.0%
**Linearity** under 35% noise	**Charge range (pC)**	**Linearity error = ε_l_ = (Max** **ε** ** _q_ ** **value–Min** **ε** ** _q_ ** **value)/2** **Requirement ε_l_ ≤ 5% under 35% noise level**			
	50 to 2.400	±0.3%	±0.6%	±13.2%	±3.6%
	10 to 2.400	±0.6%	±12.7%	±14.0%	±4.1%
**PD time (*T_PD_*)** under 35% noise q = 200 Pc	***T_PD_*** **range (ns)**	** $T_{\mathrm{PD}}\;\mathrm{error}=\varepsilon_{T_{PD}}$ ** **= (Max** **ε** ** _q_ ** **value–Min** **ε** ** _q_ ** **value)/2** ** $\mathrm{Requirement}\;\varepsilon_{T_{PD}}$ ** **≤ 30% under 35% noise level**			
	37.5–150	±25.1%	±2.0% (*)	±2.8%	±25.4%
	8 to 150	±29.1%	Not applicable (**)	±4.0%	±29.6%
**Resolution time** under 35% noise q = 200 pC	***t_res_*** **range (μs)**	**PD repetition rate error** **Requirement Max** **|*****ε_n_*****|**** ≤ 2% under 35% noise level, at least 20 μs**			
	2.500–10	10 μs, *ε_n_* = −0.4%	20 μs, *ε_n_* = 0.0%	2.5 ms, ε_n_ = 0.0%	10 μs, *ε_n_* = 0.0%

(*) In the PD time test, the direct readings were corrected using Equation (3) considering the nominal *T_PD_* values. For this reason, the results obtained with this analyser cannot be considered as independent measurements. (**) They are not applicable because the digital recorder used a sampling rate of 10 ns.

**Table 5 sensors-23-06317-t005:** Metrological tests results of PD analysers operating for continuous PD monitoring.

PD Analyser for Continuous PD Monitoring		PD Analyser B-1	PD Analyser B-2	PD Analyser B-3	PD Analyser C
HFCT bandwidth f_1_–f_2_ (MHz)		0.2–20	0.08–61	0.004–1.100	0.2–20
Charge measurement method f_1_–f_2_ (MHz)		Quasi-integration f_1_ = 1.0, f_2_ = 4.0	Applying formula (3)	Integration in time domain	Quasi-integration f_1_ = 2.45, f_2_ = 3.95
Digitiser: Bandwidth (Sampling rate)		30 MHz (60 MS/s)	50 MHz (100 MS/s)	20 MHz (1.25 GS/s)	50 MHz (100 MS/s)
Filtering method f_1_–f_2_ (MHz)		Passband filter Manual selection of f_1_, f_2_ and trigger level f_1_ = 1.75, f_2_ = 3.25	4th-order passband filter Butterworth f_1_ = 0.05, f_2_ = 45 (software)	8th-order lowpass filter f_2_ = 20 (hardware)	Automatic wavelet filter for recognising PD pulses and pulsating noises
Metrological tests					
**Noise** **rejection** q = 100 pC	**Noise** **amplitude (%)**	**Charge error and PD repetition rate errors versus noise level** **Requirements** **Max** **|****ε_q_****|** **≤ 30%, Max** **|****ε_n_****|** **≤ 2% under 200% noise level**			
	200	ε_q_ = 4.1%	ε_q_ = −0.4%	ε_q_ = −40.0%	ε_q_ = −11.7%
		ε_n_ = 0.0%	ε_n_ = 0.0%	ε_n_ = 0.7%	ε_n_ = −0.0%
	100	ε_q_ = 0.8%	ε_q_ = 0.0%	ε_q_ = 16.5%	ε_q_ = −1.6%
		ε_n_ = 0.0%	ε_n_ = 0.0%	ε_n_ = 0.7%	ε_n_ = −0.0%
	50	ε_q_ = −0.1%	ε_q_ = 0.1%	ε_q_ = 8.7%	ε_q_ = 1.7%
		ε_n_ = 0.0%	ε_n_ = 0.0%	ε_n_ = 0.0%	ε_n_ = −0.0%
	20	ε_q_ = 0.0%	ε_q_ = 0.2%	ε_q_ = 3.5%	ε_q_ = 1.7%
		ε_n_ = 0.0%	ε_n_ = 0.2%	ε_n_ = 0.0%	ε_n_ = −0.0%
**Linearity** under noise	**Charge range q (pC)**	**Linearity error = ε_l_ = (Max** **ε** ** _q_ ** **value–Min** **ε** ** _q_ ** **value)/2** **Requirement ε_l_ ≤ 5% under 10% noise level**			
	50 to 2.400	±0.1%	±0.5%	±2.0%	±0.2%
	10 to 2.400	±0.7%	±11.0%	±85.0%	±0.8%
**PD time (*T_PD_*)** under noise q = 200 pC	**T_PD_ range (ns)**	** $T_{\mathrm{PD}}\;\mathrm{error}=\varepsilon_{T_{PD}}$ ** **= (Max** **ε** ** _q_ ** **value–Min** **ε** ** _q_ ** **value)/2** ** $\mathrm{Requirement}\;\varepsilon_{T_{PD}}$ ** **≤ 30% under 10% noise level**			
	37.5 to 150	±24.1%	± 2.1% (*)	±0.8%	±25.5%
	8 to 150	±28.7%	Not applicable (**)	±1.5%	±29.8%
**Resolution time** under noise q = 200 pC	** *t_res_* ** **range (μs)**	**PD repetition rate error** **Requirement Max** **|** ** *ε_n_* ** **|** **≤ 2% under 10% noise level, at least 20 μs**			
	2.500–10	10 μs, *ε_n_* = 0.0%	20 μs, *ε_n_* = 0.0%	2.5 ms, *ε_n_* = 0.5%	10 μs, *ε_n_* = 0.0%

(*) In the PD time test, the direct readings were corrected using Equation (3) considering the nominal *T_PD_* values. For this reason, the results obtained with this analyser cannot be considered as independent measurements. (**) They are not applicable because the digital recorder used a sampling rate of 10 ns.

## Data Availability

Data is unavailable due to privacy restrictions.

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
