# Peer review of "Metrological Qualification of PD Analysers for Insulation Diagnosis of HVDC and HVAC Grids"

_sensors, 2023, doi:10.3390/s23146317_

Round 1

Reviewer 1 Report

The article presents the  Metrological Qualification of PD Analysers for Insulation Diagnosis of HVDC and HVAC Cable Systems. The paper should consider for the following suggestions: 

There is much research reported and published on this title last few years and the authors need to review them in their manuscript.

The abstract is lengthy and not focused. For research papers, authors must mention the most important numerical findings, the novelty of the work and the specific method that they have implemented in their article. Please shorten the abstract and more focus on the findings and novelty.

Does the ferroresonance effect is considered and analyzed?

Please update the list of references with the recent year's research.

Author Response

ALL THE CORRECTIONS INCLUDED IN THE TEXT OF THE PAPER ARE WRITTEN IN BLUE COLOR.

To better focus the topic of the article, "cable systems" has been replaced by "grids" in the text.

ANSWERS FOR REVIEWER 1

  1. There is much research reported and published on this title last few years and the authors need to review them in their manuscript.

Thank you very much for this indication. The paper has been completed with the following references recently published.

[1]          Sikorski, W.; Wielewski, A. Low-Cost Online Partial Discharge Monitoring System for Power Transformers. Sensors 2023, 23, 3405. 

[5]          Bassan, F.R.; Rosolem, J.B.; Floridia, C.; Penze, R.S.; Aires, B.N.; Roncolatto, R.A.; Peres, R.; Júnior, J.R.N.; Fracarolli, J.P.V.; da Costa, E.F.; Cardoso, F.H.; Pereira, F.R.; Furoni, C.C.; Coimbra, C.M.; Riboldi, V.B.; Omae, C.; de Moraes, M. Multi-Parameter Optical Monitoring Solution Applied to Underground Medium-Voltage Electric Power Distribution Networks. Sensors 2023, 23, 5066.

[6]          Hu, X.; Zhang, G.; Liu, X.; Chen, K.; Zhang, X. Design of High-Sensitivity Flexible Low-Profile Spiral Antenna Sensor for GIS Built-in PD Detection. Sensors 2023, 23, 4722.

[7]          Riba, J.-R. Application of Image Sensors to Detect and Locate Electrical Discharges: A Review. Sensors 2022, 22, 5886.

[12]      M. A. Fard, M. E. Farrag, S. G. McMeekin and A. J. Reid, "Partial discharge behavior under operational and anomalous conditions in HVDC systems," in IEEE Transactions on Dielectrics and Electrical Insulation, vol. 24, no. 3, pp. 1494-1502, June 2017.

[20]       Wang, Y.; Chen, P.; Zhao, Y.; Sun, Y. A Denoising Method for Mining Cable PD Signal Based on Genetic Algorithm Optimization of VMD and Wavelet Threshold. Sensors 2022, 22, 9386.

  1. The abstract is lengthy and not focused. For research papers, authors must mention the most important numerical findings, the novelty of the work and the specific method that they have implemented in their article. Please shorten the abstract and more focus on the findings and novelty.

Thank you for this indication, now the abstract has been modified, shortened and improved as follows.

NEW ABSTRACT

Abstract: On-site partial discharge (PD) measurements have turned out to be a very efficient technique to determine the insulation condition in high-voltage electrical grids (AIS, cable systems, GIS, HVDC converters, etc.), however, there is not any standardized procedure to determine the performances of PD measuring systems. In on-line and on-site PD measurements, high frequency current transformers (HFCTs) are commonly used as sensors, as they allow monitoring over long distances in high-voltage installations. To ensure the required performances, the metrological qualification of the PD analysers is necessary applying an evaluation procedure. A novel evaluation procedure has been established to specify the quantities to be measured (electrical charge and PD repetition rate) and to describe the evaluation tests considering the measuring influence parameters: noise, charge amplitude, pulse width, and time interval between consecutive pulses. This procedure has been applied to different types of PD analysers used for off-line measurements, sporadic on-line measurements and continuous PD monitoring. The procedure was validated in a round robin test involving two metrological institutes (RISE from Sweden and FFII from Spain) and three universities (TUDelft from the Netherlands, TAU from Finland and UPM from Spain). With this round robin test the effectiveness of the proposed qualification procedure for discriminating between efficient and inappropriate PD analysers has been demonstrated. Furthermore, it has been shown that the PD charge quantity can be properly determined for on-line measurements and continuous monitoring, by integrating the pulse signals acquired with HFCT sensors. In this case, these sensors must have a flat frequency spectrum in the range between several tens of kHz and at least two tens of MHz, where the frequency pulse content is more significant. The proposed qualification procedure can be useful for improving the future versions of the technical specification TS IEC 62478 and the standard IEC 60270.

Previous abstract. For the achievement of adequate quality standards in high-voltage (HV) AC and DC transmission and distribution lines, the implementation of routine dielectric condition tests and monitoring systems is highly recommended. In this sense, partial discharge (PD) measurement has turned out to be a very efficient and cost-effective technique for improving the condition of HV power grids. The charge of the PD pulses generated in the insulation defects of cable systems remain nearly constant when they travel along the cable sheaths. However, their detection far from the defect location is difficult because their amplitude is strongly attenuated. In monitoring applications, the use of high frequency current transformers (HFCT) sensors for PD activity detection is an effective way to supervise long distances in HV facilities. These sensors are installed in the assets grounding systems, as for example in the cable sheaths earth connections. To ensure the realization of accurate diagnosis, the metrological qualification of the PD analysers is necessary. With the research presented in this paper, measurement errors caused by PD analysers due to various influence parameters are determined by performing a set of four metrological tests: noise rejection, linearity, pulse width influence and resolution time. For this metrological qualification of the PD analysers a testing procedure has been developed and applied in a European round robin test, in which two metrological institutes RISE and FFII and three universities TUDelft, TAU and UPM have participated. For the practical validation of the qualification procedure, various types of PD analysers were tested. Furthermore, the different methods applied for the charge value determination are reviewed and discussed to promote the development of better PD analysers. This research has been performed in the framework of the project Future Energy 19ENG02 of EURAMET and the results obtained will be useful to support the development of the future versions of TS IEC 62478 and IEC 60270.

  1. Does the ferroresonance effect is considered and analysed?

No, as this phenomenon is not related with the transient events associated with the partial discharges generated in insulation defects.

  1. Please update the list of references with the recent year´s research.

The new references are indicated in point 1.

Reviewer 2 Report

The paper presents the study about metrological qualification of partial discharges analyzers for insulation diagnosis of DC and AC high voltage cables. Authors determined measurement errors caused by partial discharges analyzers due to various influence parameters by performing a set of metrological tests. For this metrological qualification of the analyzers a testing procedure was developed. Different types of partial discharges were tested. Different methods applied for the charge value determination were reviewed and discussed to promote the development of better analyzers.

Dear author, thank you very much for interesting paper about PD analyzers dedicated to AC and DC high voltage cables. I put some comments and questions.

Comments:

1. line 27, it should be “PD”, not “DP”. DP means Degree of Polymerization of cellulose materials, used as high voltage insulation in cable and power transformers. Please correct. See also line 307, there is the same mistake. Please correct too.

1. The introduction is well organized. Authors could add some important samples of defects in cable, which generate partial discharges.

3. formulas 1-3, each parameter of the formulas should be described in the text, such as alpha and betta. Please complete.

4. did authors use some sample of cable to test and verify your analyzers. If yes, please describe it.

5. I think, some fundamental problems of measurement of PD should be described in the beginning of the paper, what would better motivate made investigations.

Author Response

ALL THE CORRECTIONS INCLUDED IN THE TEXT OF THE PAPER ARE WRITTEN IN BLUE COLOR.

ANSWERS FOR REVIEWER 2

  1. Line 27, it should be “PD”, not “DP”, DP means Degree of Polymerization of cellulose materials, used as high voltage insulation in cable and power transformers. Please correct. See also line 307, there is the same mistake. Please correct too.

Thank you. This has been already corrected.

  1. The introduction is well organized. Authors could add some important samples of defects in cable, which generate partial discharges.

Thank you for this indication. We have added the following test in the introduction.

… have proven to be very useful to detect insulation defects in HVAC installations [1-7]. The most critical insulation defects that generate PD activity in electrical grids are internal or void type defects and internal surface defects. Less consideration should be given to external surface defects or to floating potential defects and when the corona effect is detected. Among …

  1. Formulas 1-3, each parameter of the formulas should be described in the text, such as alpha and betta. Please complete.

Very good indication. We have completed the text as follows for the better understanding of these formulas.

The waveform of the pulses generated in the PD trains corresponds to an inverse double exponential function (IDE) according to equation (1).     

                        (1)

Where ipeak is the peak value of the current pulse i(t) and α and β two time constants. The parameter k is obtained with the following equation.

                                                       (2)

The charge value of this current pulse can be calculated by equation (3):

                     (3)

Being ip.u. the current pulse per unit.

  1. Did authors use some sample of cable to test and verify your analyzers. If yes, please describe it.

This research is focused in the metrological qualification of PD analysers and not in the qualification of their diagnostic tools, as for example those related with the location of the insulation defect where the PD are generated. For the qualification of location tools, it is necessary to use samples of cables in order to check their effectiveness for the location of the PD sources associated with the insulation defect in a certain cable system. Thus, as this research is focussed in metrological tests, the use of a sample cable is not necessary. However, the technical consideration related to the effect of the PD pulses propagation along a cable sample is of utmost importance and must be considered in the metrological tests. This consideration is addressed in this research by implementing the pulse width influence test (PD time test), where the TPD parameter has been introduced. A complementary procedure has been developed by the authors to qualify the functionality capabilities (not the metrological ones) of PD analysers. In the tests described in this procedure various cable samples have been used to perform the following diagnostic tests: PD clustering, PD location and PD recognition. The results are presented in a new paper. This paper is in this moment under review.

  1. I think, some fundamental problems of measurement of PD should be described in the beginning of the paper, what would better motivate made investigations.

Thank you for this comment. We have included the following text in the Introduction. In our opinion, this text reinforce the importance of the research developed.

In on-site PD tests, special attention must be paid in the influence factors that affect to the measurements. These influence factors can be summarized as follows: the background noise, the PD magnitude variability, the PD pulses width and the minimum time interval between consecutive pulses. The previous influence factors have been considered in the set of four tests proposed in this research which are described in section 5.
